# Unique, Specific CART Receptor-Independent Regulatory Mechanism of CART(55-102) Peptide in Spinal Nociceptive Transmission and Its Relation to Dipeptidyl-Peptidase 4 (DDP4)

**DOI:** 10.3390/ijms24020918

**Published:** 2023-01-04

**Authors:** Márk Kozsurek, Kornél Király, Klára Gyimesi, Erika Lukácsi, Csaba Fekete, Balázs Gereben, Petra Mohácsik, Zsuzsanna Helyes, Kata Bölcskei, Valéria Tékus, Károly Pap, Edina Szűcs, Sándor Benyhe, Tímea Imre, Pál Szabó, Andrea Gajtkó, Krisztina Holló, Zita Puskár

**Affiliations:** 1Department of Anatomy, Histology and Embryology, Semmelweis University, H-1094 Budapest, Hungary; 2Department of Pharmacology and Pharmacotherapy, Semmelweis University, H-1089 Budapest, Hungary; 3Department of Anaesthesiology, Uzsoki Hospital, H-1145 Budapest, Hungary; 4Laboratory of Integrative Neuroendocrinology, Institute of Experimental Medicine, Eötvös Loránd Research Network, H-1083 Budapest, Hungary; 5Department of Medicine, Division of Endocrinology, Diabetes and Metabolism, Tupper Research Institute, Tufts Medical Center, Boston, MA 02111, USA; 6Department of Pharmacology and Pharmacotherapy, University of Pécs, H-7624 Pécs, Hungary; 7National Laboratory for Drug Research and Development, H-1117 Budapest, Hungary; 8Chronic Pain Research Group, Eötvös Loránd Research Network, H-7624 Pécs, Hungary; 9Department of Orthopaedics and Traumatology, Uzsoki Hospital, H-1145 Budapest, Hungary; 10Institute of Biochemistry, Biological Research Centre, Eötvös Loránd Research Network, H-6726 Szeged, Hungary; 11MS Metabolomics Laboratory, Instrumentation Centre, Research Centre for Natural Sciences, Eötvös Loránd Research Network, H-1117 Budapest, Hungary; 12Department of Anatomy, Histology and Embryology, Faculty of Medicine, University of Debrecen, H-4032 Debrecen, Hungary

**Keywords:** cocaine- and amphetamine-regulated transcript (CART) peptide, dipeptidyl-peptidase-4 (DPP4), chronic pain, hyperalgesia, allodynia, spinal cord

## Abstract

Cocaine- and amphetamine-regulated transcript (CART) peptides are involved in several physiological and pathological processes, but their mechanism of action is unrevealed due to the lack of identified receptor(s). We provided evidence for the antihyperalgesic effect of CART(55-102) by inhibiting dipeptidyl-peptidase 4 (DPP4) in astrocytes and consequently reducing neuroinflammation in the rat spinal dorsal horn in a carrageenan-evoked inflammation model. Both naturally occurring CART(55-102) and CART(62-102) peptides are present in the spinal cord. CART(55-102) is not involved in acute nociception but regulates spinal pain transmission during peripheral inflammation. While the full-length peptide with a globular motif contributes to hyperalgesia, its N-terminal inhibits this process. Although the anti-hyperalgesic effects of CART(55-102), CART(55-76), and CART(62-76) are blocked by opioid receptor antagonists in our inflammatory models, but not in neuropathic Seltzer model, none of them bind to any opioid or G-protein coupled receptors. DPP4 interacts with Toll-like receptor 4 (TLR4) signalling in spinal astrocytes and enhances the TLR4-induced expression of interleukin-6 and tumour necrosis factor alpha contributing to inflammatory pain. Depending on the state of inflammation, CART(55-102) is processed in the spinal cord, resulting in the generation of biologically active isoleucine-proline-isoleucine (IPI) tripeptide, which inhibits DPP4, leading to significantly decreased glia-derived cytokine production and hyperalgesia.

## 1. Introduction

The discovery of the cocaine- and amphetamine-regulated transcript (CART) peptide family [1,2] raised high hopes of a potential therapy for obesity due to its extreme anorexigenic effect [3]. Later, CART peptides were implicated in several physiological and pathological processes [4,5,6]. The chemical structures of the naturally occurring CART(55-102) and CART(62-102) peptides were described [7], and specific binding sites were demonstrated in different cell types [8,9,10,11]. However, despite extensive research in the past two decades, specific CART receptors have not been identified and cloned yet. Studies focusing on the structure–activity relationship of CART fragments (Figure 1) revealed that the full-length peptide and the disulphide-bridges in the globular region are required for both receptor binding and the CART-induced regulation of food intake [12,13,14].

In addition to brain regions regulating food intake/energy balance, massive CART-containing innervation exists in the superficial dorsal horn of the spinal cord [15,16,17]. CART peptide is present in the majority of peptidergic nociceptive C-fibres terminating on lamina I projection neurons [18,19], suggesting the involvement of CART in nociceptive information processing. Exogenous CART(55-102) enhances NMDA-induced hyperalgesia [20], potentiates the analgesic action of morphine in acute nociception [21], and attenuates hyperalgesia in neuropathy [22]. These effects mark multiple signalling pathways of CART peptides forecasting mechanisms of actions that do not require a specific receptor binding. Since the first three amino acids of the N-terminal region of CART(55-102) are isoleucine-proline-isoleucine (IPI), an inhibitor of the dipeptidyl-peptidase 4 (DPP4) enzyme [23,24], we assumed that CART(55-102) may regulate spinal nociceptive transmission by acting as a DPP4 inhibitor.

DPP4 is a membrane glycoprotein expressed in many cell types, but it is also present in soluble form in body fluids. As a serine protease, DPP4 cleaves dipeptides from oligopeptides or proteins containing proline/alanine in the penultimate position, resulting in their altered biological activity. According to its function in the regulation of postprandial insulin secretion, DPP4 has become one of the major targets for the therapy of type II diabetes [25,26,27]. DPP4 also possesses non-catalytic functions in T-lymphocyte activation or inflammation [28,29] and acts as a functional receptor for the Middle East Respiratory Syndrome coronavirus, and its involvement in COVID-19 has been proposed [30,31,32]. DPP4 is present in the spinal cord, and its glial expression is upregulated during chronic pain. Furthermore, the spinal application of DPP4 inhibitors has a robust “µ-opioid receptor mediated” antihyperalgesic effect on inflammation and also diminishes neuropathic hyperalgesia [33].

Long-term potentiation (LTP) in C-fibre synapses is the cellular model of pain amplification (hyperalgesia). Hyperalgesia and LTP can be induced in an activity dependant manner, but activity independent forms also exist [34]. A fundamentally new type of LTP called gliogenic LTP was discovered recently. It is induced by glial cells and is mediated by extracellular messengers, including tumour necrosis factor alpha (TNFα) [35]. Soluble DPP4 enhances interleukin (IL)-6 and the TNFα expression of monocytes induced by bacterial lipopolysaccharides (LPS) [36], suggesting that spinal DPP4 may also contribute to the development of gliogenic LTP and hyperalgesia by stimulating cytokine production.

Therefore, in the present study, we tested the hypothesis that CART(55-102) regulates pain processing as a DPP4 inhibitor, decreasing glial cytokine expression, and we determined whether this action is independent from the previously suggested receptor-mediated mechanism of CART signalling via a specific CART receptor.

## 2. Results

### 2.1. Coexistence of CART and DPP4 in the Spinal Nociceptive Pathways

Multiple immunofluorescent labelling was designed to look for anatomical evidence of the possible interaction between CART peptides and DPP4. We showed previously [19] that the majority of CART peptides in the spinal cord originated from calcitonin gene-related peptide (CGRP) and/or substance P (SP) containing nociceptive primary afferents, but SP immunoreactive interneurons also expressed the peptide. In this study, our multiple immunofluorescent labelling with CART, DPP4, and SP or CGRP antibodies revealed that the DPP4 enzyme existed on CART immunoreactive axon terminals and in the close proximity of CART-ergic boutons (Figure 2). The close proximity or co-localisation of CART peptide and DPP4 in nociceptive fibres supported the hypothesized interaction.

### 2.2. Presence of Both CART(55-102) and CART(62-102) in the Spinal Cord

As available CART antibodies that were raised against the C-terminal portion of the CART peptides, which is identical in both CART(55-102) and CART(62-102), consequently recognise both forms of CART, this does not allow to distinguish between the presence of CART(55-102) and CART(62-102) with immunohistochemistry. Therefore, western blot analyses were performed to determine the presence of the potential IPI donor CART(55-102) in the spinal cord. Western blot demonstrated that both the 5.2 and 4.5 kDa CART fragments, corresponding to CART(55-102) and CART(62-102), respectively, were present in the spinal cord (Appendix A).

### 2.3. Effects of CART Peptides in Acute Nociceptive, Inflammatory, and Neuropathic Pain States

In order to compare the pharmacological effect of CART peptides and IPI, the established DPP4 inhibitor, we tested CART fragments in an acute nociceptive tail-flick test and in carrageenan-induced subacute inflammatory hyperalgesia and partial nerve ligation-induced neuropathic pain models, in which IPI action was previously measured [33].

Intrathecal (i.t.) administration of 0.1–0.3 nmol/rat CART(55-102) significantly decreased the mechanical hyperalgesia in the carrageenan-induced acute inflammation model (Figure 3), while i.t. 0.1–1.0 nmol/rat CART(62-102) had no antihyperalgesic effect The i.t. administration of more than 0.3 nmol/rat CART(55-102) and 1.0 nmol/rat CART(62-102), however, caused abnormal motor perturbances (ascending tremor) that prevented precise measurements. This effect also hampered the nociceptive tail-flick test.

Considering the in vivo activity of different CART peptide fragments [37], we hypothesized that the globular region of the peptide and the suggested specific CART receptor binding may have been responsible for the motor disturbance; thus, CART(55-76) lacking this globular motif (Figure 1) was used in the next experiments.

Intrathecal administration of 0.1–3.0 nmol/rat CART(55-76) had a significant antihyperalgesic effect on carrageenan-induced inflammation (Figure 4A,B), while i.t. administration of even 10 nmol/rat CART(55-76) had no effect on the acute nociceptive tail-flick test (Appendix A) and did not cause any motor side effect. These data proved that the impact of CART(55-102) on antihyperalgesia was mediated by different mechanisms than the effects of CART(55-102) on the motor system or on the regulation of food intake that requires the globular domain of CART(55-102).

Since the antihyperalgesic effect of IPI was shown to be “µ-opioid receptor (MOR)” mediated in carrageenan-induced inflammation, the relationship between CART(55-76) and the opioid system was also tested. The antihyperalgesic action of CART(55-76) was fully blocked by the non-specific opioid receptor antagonist naltrexone hydrochloride (NTX, Figure 4C) and the selective MOR antagonist D-Phe-Cys-Tyr-D-Trp-Arg-Thr-Pen-Thr-NH2 (CTAP, Figure 4D). Neither the δ opioid receptor (DOR) selective antagonist H-Tyr-Tic(CH2NH)-Phe-Phe-OH (TIPP[Ψ], Figure 4E) nor the κ opioid receptor (KOR) selective antagonist 5′-guanidinonaltrindole (gNTI, Figure 4F) had an influence on the antihyperalgesic effect of CART(55-76). CART(55-76) never altered the nociceptive threshold of the contralateral hind paw significantly, and none of the antagonists tested had a significant effect when they were injected alone.

These data revealed a novel mechanism of action for CART(55-102) peptide and highlighted the importance of the N-terminal region of CART(55-102) and CART(55-76) in the regulation of nociceptive processes. To further test the significance of the N-terminal portion of the CART(55-76) fragment, CART(62-76) lacking the IPI was administered i.t. in the acute nociceptive tail-flick test and in the carrageenan-induced hyperalgesia model. CART(62-76) did not show any effect in the acute nociceptive test (Appendix A) but produced an antihyperalgesic effect in acute inflammation. As the dose-response curves demonstrated, the antihyperalgesic effect of CART(62-76) appeared in a much higher concentration than in the case of CART(55-76) (Figure 5A,B). Although NTX fully blocked this action (Figure 5C), the µ-antagonist CTAP only slightly decreased this action (Figure 5D). The selective DOR antagonist TIPP(Ψ) did not affect (Figure 5E), but the selective KOR antagonist gNTI completely blocked the antihyperalgesic effect of CART(62-76) (Figure 5F), indicating the involvement of different receptor types or mechanisms.

CART(55-76) exerted an antihyperalgesic effect in the neuropathic pain model, but it was found to be “non-opioid”-dependent (Appendix A). Surprisingly, CART(62-76) was ineffective alone, but when co-administered with NTX, an antihyperalgesic effect was observed (Appendix A). None of the peptides altered either mechanical (Appendix A) or cold allodynia (Appendix A) developed after partial sciatic nerve ligation. Since, as stated previously, the full-length peptide and the disulphide bridges in the globular region of naturally occurring CART(55-102) and CART(62-102) are necessary for receptor binding, these results proved that the N-terminal region of both cases can act in a specific CART receptor-independent manner. Furthermore, pharmacological features of CART peptides containing N-terminal IPI CART(55-102) or CART(55-76), but not the shorter fragments CART(62-102) or CART(62-76), are fairly similar to pharmacological features of the DPP4 enzyme inhibitor IPI.

### 2.4. CART Peptides and Opioid Receptor Binding

The opioid-mediated effects of CART(55-76) and CART(62-76) raised the possibility that these peptides could bind to opioid receptors. To test this, different binding assays were carried out. The peptides CART(55-76) and CART(62-76) did not bind to MOR and KOR in competition binding experiments up to a concentration of 10 μM. Similarly, no GTP-protein activation was observed in the functional [^35^S]GTPγS binding assay (Appendix A). These experiments demonstrated that none of the CART fragments showed an affinity for binding to any type of opioid or G protein-coupled receptor.

### 2.5. Effects of CART(55-76) on the Activity of the DPP4 Enzyme

The first three amino acids of CART(55-102) and CART(55-76) are IPI, which suggests that the peptides may work as a DPP4 inhibitor. To test this possibility, a DPP4 inhibitory screening assay was carried out to compare the effect of the reference DPP4 inhibitor sitagliptin, IPI, and CART(55-76) on enzyme activity. Sitagliptin completely abolished DPP4 enzyme activity, while IPI decreased the activity to zero in a concentration dependent manner. CART(55-76) had no inhibitory effect at all (Appendix A), suggesting that the peptide in this chemical form was not a DPP4 inhibitor.

### 2.6. IPI Is Cleaved from CART(55-76) in the Rat Spinal Cord in Carrageenan-Induced Hind Paw Inflamation

The extreme similarity of the pharmacological properties of IPI and CART(55-76) in different pain models suggested that the IPI may be cleaved from the CART(55-76) in certain conditions. To test this option, cerebrospinal fluid (CSF) and spinal segments taken from naïve, inflamed, and CART(55-76)-treated inflamed rats were processed for mass spectrometry to measure their IPI content. Analysis of the CSF samples showed no detectable IPI level in any experimental groups. Although the IPI content did not differ significantly in naïve and inflamed spinal segments, its concentration increased significantly in the CART(55-76)-treated inflamed animals, suggesting that IPI was cleaved immediately from exogenously applied CART(55-76) peptides under inflammatory conditions (Figure 6).

### 2.7. Effects of DPP4 Inhibitor IPI on IL-6 and TNFα Expression of Spinal Astrocytes

We demonstrated previously that, in the spinal cord, microglia and astrocytes expressed DPP4 protein with one and two orders of magnitude higher than neurons, respectively, and DPP4 significantly increased in astrocytes during carrageenan-induced peripheral inflammation and in microglia in neuropathy [33]. To test the hypothesis that membrane-bound DPP4 contributes to the enhanced TNFα and IL-6 production of astrocytes, spinal astrocytes were cultured. In immunocytochemical experiments, astrocytes were labelled with antibodies raised against the glial fibrillary acidic protein (GFAP). In the control cultures, very low DPP4 immunoreactivity was detected (Figure 7(A1–A3)). However, when the cultures were activated by 1 μg/mL LPS [38], the intensity of DPP4 staining considerably increased upon 6 h of treatment (Figure 7(B1–B3)). Significantly increased DPP4 expression was also confirmed after 16 h of LPS treatment by western blot experiments (Figure 7C). The DPP4 expression level increased by 74.3 ± 9.6% (from 0.62 ± 0.084 optical density value obtained for the control cultures to 1.08 ± 0.059 value calculated for the LPS stimulated cultures) (Figure 7D).

After detecting enhanced expression of DPP4 in LPS-treated cultures, we tested the expression of two pro-inflammatory cytokines in the supernatants of the astrocyte cultures. We observed that LPS treatment significantly enhanced the TNFα and IL-6 production of the spinal astrocyte cultures (Figure 8). The highest increase was detected in the case of LPS-induced IL-6 production (882 ± 14%), but LPS-induced TNFα secretion was also considerably enhanced (368 ± 9%). Then, to confirm the possible role of DPP4 in the regulation of LPS-induced cytokine secretion of spinal astrocytes, the cell cultures were treated with the DPP4 inhibitor IPI tripeptide in 0.1 and 1 μM concentrations. The relative expression of LPS-induced IL-6 was reduced to 469.6 ± 4.5% (*p* = 0.000231) and the LPS-induced TNFα secretion returned to the control level (94.9 ± 10%, *p* = 0.00139).

LPS is a Toll-like receptor 4 (TLR4) agonist, and our pilot experiments showed that the intraplantar injection of carrageenan induces TLR4 and caveolin-1 (CAV1) expression in the spinal cord (unpublished observation). Since data about astrocytic TLR4 expression in the literature were controversial, we tested the presence of TLR4 in the spinal astrocyte cultures together with CAV1 expression. CAV1 was shown earlier to bind CD26/DPP4 [39]. In control cultures, we observed even distribution of the two markers in the cytoplasmic compartment of the GFAP positive astrocytes (Figure 7(E1–E4)). In the case of the LPS-stimulated cultures, a characteristic rearrangement of the subcellular localization of the two markers was shown, and both proteins were trafficked toward the cell surface (Figure 7(B1–B4)), which can be a sign of TLR4 activation [40]. Our experiments suggest that spinal astrocytes increase DPP4 expression in response to TLR4 activation, which then contributes to enhanced cytokine production.

## 3. Discussion

The main findings of this study are that (1) both CART(55-102) and CART(62-102) are present in the spinal dorsal horn and co-exist with DPP4 in lamina I and II; (2) CART(55-102) does not alter the acute nociceptive threshold but shows an antihyperalgesic effect in carrageenan-induced inflammation; (3) both CART peptides with the globular motif provoke an ascending tremor in a dose-dependent manner; (4) the antihyperalgesic effects of CART(55-76) and CART(62-76) are blocked by opioid receptor antagonists in inflammatory pain model; (5) none of the CART(55-76) and CART(62-76) peptides bind to any opioid or G-protein coupled receptors; (6) in the neuropathic model, the antihyperalgesic effect of CART(55-76) is not eliminated by opioid receptor antagonists, while CART(62-76) has no antihyperalgesic effect at all; (7) IPI tripeptide with DPP4 inhibitory activity is cleaved from CART(55-76) in the spinal cord during peripheral inflammation; (8) TLR4 agonist LPS provokes the upregulation of DPP4 and the trafficking of DPP4, TLR4, and CAV1 toward the cell surface in astrocytes, (9) IPI significantly decreases the LPS-induced proinflammatory cytokine (TNFα and IL6) release of this cell type.

### 3.1. CART Peptide in the Spinal Nociceptive System

Our results supported by the literature suggest a very complex role of CART(55-102) in pain transmission. CART(55-102) may not have a key role in acute nociception, but the full-length CART(55-102), including the globular region released from nociceptive afferents, can contribute to the development of hyperalgesia through a specific CART receptor mediated effect in different prolonged pain states [20]. When IPI is cleaved from CART(55-102), binding to DPP4 can induce the termination of hyperalgesia [33]. This newly discovered regulatory mechanism may help to develop novel analgesics, since the currently available DPP4 inhibitors used for diabetes therapy do not pass the blood brain barrier [41].

Neutral endopeptidase (NEP), a membrane-bound zinc-metalloprotease enzyme, can cleave IPI from CART(55-102) peptides. NEP is found in the CNS on nerve endings, activated astrocytes, and microglia, and its involvement in inflammatory pain has been suggested [42].

The coexistence of CART and DPP4 is not restricted to the spinal cord, since both of them are found in different anatomical locations such as the gastrointestinal tract, the pancreas, and blood vessels [43,44,45,46,47,48]. The regulatory interaction between CART and DPP4 is likely to be general and not restricted to the CNS.

### 3.2. CART Peptide in the Spinal Motor System

CART-ergic fibres were found not only in the superficial laminae but also in the intermediate grey matter and the motor columns of the spinal cord [19]. These regions contain the central pattern generator circuits, premotor interneurons, and motor neurons that conduct motor performance [49], suggesting a possible involvement of CART in the modulation of motor activity.

Epileptiform activity with tremors was described following the intracerebroventricular application of a high dose of CART(55-102) peptide in rats [50]. Although the exact neuronal mechanisms underlying this finding were unclear, the involvement of the dopaminergic system was suggested. In our experiments, intrathecally applied CART(55-102) and CART(62-102) peptides caused a robust ascending tremor in a dose-dependent manner, which started in the tail, spread to the lower limbs, trunk, upper limbs, and to the head, suggesting a spinal mechanism instead of central effects. Furthermore, the tremor did not appear if CART peptides lacked the globular motif. Fitting well with our results, the tremor was also observed in goldfish (*Carassius auratus*) following the intracerebroventricular application of a high dose of CART(55-102) but not CART(62-76) [51]. These observations support the importance of the globular motif and the proposed CART receptor in the development of physiological or pathological tremors.

### 3.3. Contribution of DPP4 to Hyperalgesia Development

We demonstrated the existence of DPP4 on glial cells, neuronal cell bodies, and axon terminals including CART-containing nociceptive afferents in the spinal cord. Microglia and astrocytes expressed DPP4 with one and two orders of magnitude higher than neurons, respectively, and DPP4 expression significantly increased in astrocytes during carrageenan-induced peripheral inflammation and in microglia in neuropathy. Spinal administration of DPP4 inhibitor IPI significantly decreased the hyperalgesia provoked by the intraplantar injection of carrageenan or partial nerve ligation, which could be eliminated by general opioid antagonist NTX and the selective MOR antagonist CTAP in inflammation but not neuropathy [33].

The synaptic model of hyperalgesia is the LTP at C-fibre synapses [34,52]. The induction of hyperalgesia and LTP by inducible mediators including TNFα was discovered and called gliogenic LTP [35]. The involvement of cytokines such as IL-6 and IL-1β in pathological pain was also demonstrated [53,54,55]. The enhancement of the (TLR4 agonist) LPS-induced IL-6 and TNFα expression of monocytes by soluble DPP4 was demonstrated in vitro, revealing an important interaction between DPP4 and TLR4 signalling [36]. TLRs are membrane-bound pattern-recognizing receptors that detect molecular patterns of pathogens at the plasma membrane and endosomes. The stimulation of TLRs results in the activation of the canonical NF-κB pathway that is responsible for the transcriptional induction of pro-inflammatory cytokines, including IL-1, IL-6, TNFα, chemokines, and inflammatory mediators initiating inflammation or the activation of inflammatory cells [56]. Ikeda et al. demonstrated that the binding of soluble DPP4 to human monocytes and THP-1 cells (a human leukemia monocytic cell line) enhanced the LPS-induced IL6 and TNFα mRNA and protein expression and the upregulation of ERK in the cytosol and c-fos, NF-κB p65/p50, and CUX1 transcription factors in the nucleus. The increased binding of these transcription factors to the promoters and boosted transcription were also detected in this study. The DPP4 inhibitor algogliptin decreased the TLR4 activation-induced IL-6, IL-1β, and other cytokine expression [57].

TLR4 plays an important role in the recognition of carrageenan and the induction of inflammation in carrageenan-induced peripheral inflammation [58]. Mirroring the peripheral events, we found an upregulation of TLR4 and CAV1 in the spinal cord (unpublished observation), and elevated levels of IL-1β and TNFα were also reported in this condition [59]. The activation of the peripheral and spinal TLR4/NF-κB signalling pathway was also demonstrated in a rat model of chronic postsurgical pain [60]. TLR4 is found in sensory and other types of neurons and in glial cells [61]. Following LPS stimulation, the increase of DPP4 immunoreactivity and protein level, as well as the trafficking of TLR4 and CAV1 toward the cell surface, in cultured astrocytes together with elevated levels of IL-6 and TNFα in the supernatant were detected in our study. DPP4 inhibitor IPI significantly decreased the IL-6 release and reduced the TNFα level to the control level, suggesting that membrane-bound DPP4 also interacts with TLR4 signalling and contributes to the enhanced cytokine production. Recent studies demonstrated that DPP4 protein directly binds to CAV1, and CAV1 as a scaffolding protein is essential for inflammatory responses initiated by TLR4 activation [19]. DPP4-CAV1 interaction results in the phosphorylation of CAV1 and the activation of the NF-κB pathway by caveolin-bound signalling molecules that initiate pro-inflammatory cytokine expression [62,63]. Previous studies suggested that the TLR4 agonist-induced cytokine production of astrocytes is microglia-dependent [64,65]. Although our cell culture was prepared as pure astrocyte culture, we cannot exclude the possibility that a few microglia existed and contributed to the processes. However, we have to note that astrocyte heterogeneity exists across the brain and spinal cord [66]. Our astrocytes originated from the spinal cord, while other studies used astrocytes from different brain regions [64,65].

We demonstrated that the intrathecal application of the non-selective opioid antagonist NTX and the MOR-selective antagonist CTAP completely blocked the antihyperalgesic effects of both IPI [33] and CART(55-76) in our inflammatory pain model, suggesting a MOR-mediated effect. It was shown previously that naltrexone alone had no effect on inflammatory hyperalgesia [67]. The opioid-mediated effect is established by an exogenous peptide if an opioid receptor is activated or if an endogenous opioid release is induced. We showed herein that none of the CART fragments bind to opioid or G-protein coupled receptors. On the other hand, CART peptide- or CART fragment-induced endogenous opioid release has not been detected so far. It is accepted that NTX, naloxone [68,69,70], and CTAP [71] are TLR4 antagonists in the TLR4/myeloid differentiation protein-2 heterodimer, but it has been shown recently that NTX did not block the TLR4 agonist-induced cytokine production [72]. It was proposed that TLR4, CAV1, and DPP4 are located in the same lipid raft [63]; thus, our study suggests that CTAP/NTX docking to TLR4 may affect the binding of DPP4 inhibitors to the DPP4-CAV1 complex that cannot block the ongoing inflammatory processes.

Our experiments support that both the activity-dependent and cytokine-induced components of hyperalgesia develop during peripheral inflammation. Our study suggests that TLR4 activation occurs within the spinal cord and initiates increased DPP4 insertion into the cell membrane. Several endogenous molecules were proposed that can activate TLR4, but further studies are necessary to determine how TLR4 becomes activated in the CNS during peripheral inflammation. DPP4 interaction with TLR4 signalling through CAV1 results in enhanced cytokine production as a key element of the initiation of cytokine-related hyperalgesia and other inflammatory events. Taken together, the antihyperalgesic effects of both IPI and CART(55-102) may not be opioid-mediated but may be associated with the decrease of TLR4-mediated cytokine expression.

Exogenous CART(55-102) attenuates hyperalgesia also in neuropathy [22]. Similar to other DPP4 inhibitors such as IPI and vildagliptin, CART(55-76) peptide also showed antihyperalgesic effects in a neuropathic condition, but neither NTX nor CTAP had any effect in this model. Furthermore, DPP4 expression significantly increased in microglia but not in astrocytes in this circumstance. This suggests that DPP4 contributes to the development of hyperalgesia also in neuropathy, and the key player is microglia. Further studies are needed to reveal the signalling mechanism to understand this process.

### 3.4. CART(55-76) and CART(62-76) Fragments

The existence of CART fragments such as CART(55-76) and CART(62-76) has not been demonstrated in the spinal cord. Our study shows that if they are present, then they could be biologically active. Exogenous CART(62-76) inhibits carrageenan-induced hyperalgesia, which can be decreased by CTAP and eliminated by general opioid or KOR specific antagonists. Endogenous opioid peptides (except for nociceptin) share a common N-terminal tyrosine-glycine-glycine-phenylalanine amino acid sequence [73], bind to TLR4, and non-stereoselectively activate TLR4 signalling [74]. Although the first two amino acids of the N-terminal region of CART(62-76) are tyrosine-glycine, we showed that this CART fragment has no affinity to specific opioid receptors. Although the possibility of the CART(62-76)-induced release of endogenous opioid peptides cannot be excluded, supporting evidence does not exist in the literature.

The similarity of the first two amino acids in the N-terminal region of CART(62-76) to endogenous opioid peptides raises the possibility that this fragment may also bind to TLR4. Specific binding studies are necessary to answer this question. CART(62-76) alone had no effect on neuropathy related hyperalgesia, but together with NTX it resulted in an antihyperalgesic effect. The antiallodynic effect of NTX acting on TLR4 was shown in the chronic constriction injury model in rats [75], and low-dose NTX was shown to be effective in a variety of chronic pain states in humans [76]. Further studies are required to understand the basic mechanism of the hyperalgesic component of neuropathic pain and to determine the effect of NTX and CART(62-76) on this process.

Our results here provide new insights into our understanding of the complex role of CART peptides in nociceptive transmission and highlight that DPP4 inhibition in the CNS would be beneficial to reduce both inflammatory and neuropathic pain, as well as the neurological manifestations of peripheral inflammation.

## 4. Materials and Methods

### 4.1. Animals

Tail-flick withdrawal latency and paw mechanonociceptive threshold (Randall–Selitto test) measurements in the carrageenan-induced acute inflammation model were carried out on male Wistar rats weighing 170–230 g, which were received from the breeding colony of Semmelweis University. For the neuropathy model, male Wistar rats weighing 100–160 g at the beginning of the experiment were purchased from Charles River Laboratories via Innovo Ltd. (Gödöllő, Hungary) or Toxi-Coop Ltd. (Budapest, Hungary), and then they were bred and kept at the Laboratory Animal Centre of the University of Pécs.

Western blot experiments, mass spectrometry, and immunofluorescent labelling were performed on adult male Wistar rats (200–250 g) from the breeding colony of Semmelweis University.

For binding assays, male guinea pigs and Wistar rats of both sexes were used. Four male and female Wistar rats (250–300 g body weight) and 3 male guinea pigs (~700 g body weight) were housed in the local animal house of HAS-BRC (Szeged, Hungary) and in LAB-ÁLL LP (Budapest, Hungary), respectively.

Animals were kept in temperature-controlled rooms with 12 h light/12 h dark cycles, standard rodent chow, and tap water supplied ad libitum. The total number of animals as well as their suffering were minimized whenever possible.

### 4.2. Drugs

CART(55-102), CART(55-76), CART(62-76), and CART(62-102) peptides (Bachem–American Peptides Company, Bubendorf, Switzerland, Cat. No: 46-2-55, 46-2-57, 46-2-58 and 46-2-59, respectively) were dissolved in bi-distilled water and further diluted to working concentrations with sterile saline. IPI (Merck–Sigma-Aldrich, Darmstadt, Germany, I9759) stock solution was taken up in 25% (*w/v*) hydroxypropyl-beta-cylodextrin (Merck–Sigma-Aldrich, H107), and dilutions were made with sterile saline. Naltrexon hydrochloride (NTX), a non-selective opioid antagonist, was a kind gift from DuPont Pharmaceuticals (Geneva, Switzerland) and was applied subcutaneously (s.c.) in 0.5 mg/g b.w. dissolved in saline. The µ-opioid antagonist D-Phe-Cys-Tyr-D-Trp-Arg-Thr-Pen-Thr-NH_2_ (CTAP; Merck-Sigma-Aldrich, C6352; 200 pmol/rat), δ-antagonist H-Tyr-Tic(CH_2_NH)-Phe-Phe-OH (TIPP[Ψ]; Merck-Sigma-Aldrich, T7075; 1 nmol/rat), and κ-antagonist 5′-guanidinonaltrindole (gNTI; Merck-Sigma-Aldrich, G3416; 10 nmol/rat) were dissolved in distilled water. GDP and the GTP analogue GTPγS were purchased from Merk-Sigma-Aldrich. The highly selective μ-opioid receptor agonist enkephalin analogue, Tyr-D-Ala-Gly-(NMe)Phe-Gly-ol (DAMGO), was obtained from Bachem Holding AG (Bubendorf, Switzerland), and the highly potent and selective κ-opioid receptor agonist diphenethylamine derivative HS665 was kindly offered by Dr. Helmut Schmidhammer, University of Innsbruck, Austria [77]. The non-selective opioid receptor antagonist naloxone was kindly provided by the company Endo Laboratories DuPont de Nemours (Wilmington, DE, USA). Radio-labelled GTP analogue [^35^S]GTPγS (specific activity: 1000 Ci/mmol) was purchased from Hartmann Analytic (Braunschweig, Germany). [^3^H]DAMGO (specific activity: 38.8 Ci/mmol) and [^3^H]HS665 (specific activity: 13.1 Ci/mmol) were radio-labelled by the Isotope Laboratory of BRC (Szeged, Hungary) and were characterized previously [78]. The UltimaGold^TM^ MV aqueous scintillation cocktail was purchased from PerkinElmer (Boston, MA, USA).

### 4.3. Immunofluorescent Labelling

Three naïve rats were perfused transcardially in deep anesthesia (75 mg ketamine and 7.5 mg xylazine/kg b.w.). The perfusion was initiated with 4% (para)formaldehyde and completed with 4% (para)formaldehyde containing 10% sucrose. L3-L5 spinal segments were removed and immersed overnight into 20% sucrose dissolved in phosphate buffer saline (PBS). Tissue blocks were frozen with liquid nitrogen, and 50 µm thick sections were cut on a Vibratome. Endogenous peroxidase activity was blocked for 30 min with 1% hydrogen peroxide diluted in PBS, and then sections were transferred into PBS containing 5% normal horse serum (NHS) and 0.3% Triton X-100. Tissues were incubated for 72 h in a cocktail of mouse anti-CART [7], goat anti-DPP4 [33], and guinea pig anti-CGRP [19,79] or rat anti-SP antibodies [80] then were rinsed several times, and for 24 h, the mixture of anti-goat Alexa 488 (1:500; Thermo Fisher Scientific–Invitrogen, Waltham, MA, USA), anti-mouse Alexa 555 (1:500; Thermo Fisher Scientific–Invitrogen), anti-guinea pig Cy5, or anti-rat Cy5 (1:100; Jackson ImmunoResearch, Ely, UK) secondary antibodies all raised in donkey was applied. Both primary and secondary antibodies were dissolved in PBS containing 0.3% Triton X-100. Specifications of the used antibodies are detailed in Appendix A. Sections were then mounted in Vectashield (Vector Laboratories, Newark, CA, USA) and scanned on an LSM 780 laser scanning confocal microscope (Carl Zeiss, Jena, Germany).

### 4.4. Western Blot

An adult male Wistar rat was decapitated, and the spinal cord was removed and snap frozen on dry ice. The sample was homogenized in a lysis buffer containing 1% Triton-X 100, 70 mM NaCl, 35 mM HEPES, 3.5 mM MgCl_2_, 0.7 mM EDTA, and 0.1 mM ditiotreitol (DTT). After 30 min incubation on ice, homogenized materials were centrifuged at maximum speed for 15 min on 4 °C, and the clear supernatant was collected.

Tricine/SDS-PAGE was performed according to the protocol of Schägger [81] with some modifications: 80 µg protein of spinal cord extract and 0.2 ng of both the CART(55-102) and CART(62-102) peptides were boiled in Laemmli sample buffer with 100 mM DTT and resolved in an 8.6 cm × 6.7 cm poliacrilamide minigel at 30 V for 1 h followed by 90 V for 6–7 h. The gel was composed of 16.5% separating gel, thin 10% intermediate gel, and 4% stacking gel, all prepared with 49.5% T and 3% C acrylamide/bisacrylamide.

Semidry blotting to a 0.2 µm pore size PVDF membrane was performed overnight at 4 °C with 80 mA. The membrane was probed with rabbit polyclonal anti-CART IgG antibody [82] in a dilution of 1:4000 and incubated overnight at room temperature. Bands were visualized with chemiluminescent detection using HRP conjugated anti-rabbit secondary antibody in 1:5000.

### 4.5. Behavioural Experiments

Before the behavioural experiments, animals were always randomly assigned to the treatment and control groups, and during the measurements, the observers were blind of which group they were actually testing.

#### 4.5.1. Measuring Acute Thermal Nociception

The response to acute noxious stimuli was characterized with the radiant heat rat tail-flick test using the IITC Life Science #33 analgesia meter (Woodland Hills, CA, USA), where the cut-off time was set to 8 s to avoid tissue damage [83]. The tail-flick latency was measured once before and 5, 15, and 30 min after intrathecal (i.t.) drug application performed with a Hamilton syringe equipped with a 23-Ga needle in 5 μL volume through the L5-6 intervertebral space [67,83,84]. CART(55-76) and CART(62-76) were tested in a dose of 10 nmol/rat on 5 and 7 rats, respectively, and 6 animals were used as controls. For statistical analysis, two-way repeated measures ANOVA with Bonferroni’s post hoc test were used, and *p* < 0.05 was considered significant.

#### 4.5.2. Determining Mechanical Hyperalgesia in Carrageenan-Induced Inflammation

The mechanonociceptive threshold of the paw to pressure was determined by the Randall–Selitto method [67,83,85] using a type 37,215 Ugo Basile Analgesy-Meter (Comerio, Italy). Rats were lightly restrained, and an increasing force was applied on their paws inserted between the clamps of the apparatus. Actual force at the moment of paw withdrawal was recorded as the mechanonociceptive threshold. After two days of handling, the baseline pressure nociceptive threshold on the day of the experiment was determined on both hind paws first (–5 min), then 100 μL of l% λ-carrageenan (Merck-Sigma-Aldrich, 22049) was injected into the right hind paw (0 min) to induce an acute inflammation and the consequential drop of nociceptive thresholds (hyperalgesia). In a set of experiments, NTX or saline was applied s.c. at 165 min. The nociceptive threshold was measured again at 180 min, and the i.t. injection of solvents or drugs/drug combinations was performed. Nociceptive threshold readings were repeated at 185, 195, 210, and 240 min.

Percent antihyperalgesic effects were calculated according to the following equation: MPE (%) = 100 × (ipsilateral threshold 30 min after i.t. CART peptide application or i.t. CART fragment and opioid antagonist co-administration − hyperalgesic baseline)/(contralateral threshold at the same time − hyperalgesic baseline); hyperalgesic baseline is defined as the nociceptive threshold of the inflamed hind paw 180 min after intraplantar carrageenan injection [67,83]. Five rats were involved in each vehicle group, while 7–9 animals were used for each drug and drug combination/concentration. Time-matching data sets on different ipsilateral curves were compared with two-way repeated measures, ANOVA followed by Bonferroni post hoc test.

#### 4.5.3. Assessing Chronic Neuropathic Pain Parameters in the Partial Sciatic Nerve Ligation Model of Seltzer

Tight ligation of 1/3–1/2 of the sciatic nerve ligation induced chronic neuropathic pain which, was reflected in mechanical hyperalgesia (decrease of the threshold of a basically painful stimulus pressure) together with mechanical and cold allodynia (non-painful stimuli touch, non-noxious cold becomes painful) [86,87]. The baseline nociceptive thresholds were determined on two consecutive days using three different methods, including dynamic plantar aestesiometry (DPA; touch sensitivity; for determining mechanical allodynia), the Randall–Selitto test (mechanical hyperalgesia) and cold stimulation (cold allodynia). In 50 mg/kg i.p. pentobarbital sodium-induced deep anaesthesia (Euthasol, Produlab Pharma, Raamsdonksveer, The Netherlands), the sciatic nerves of the rats (*n* = 60) were tightly ligated high on the thigh unilaterally using a braided silk suture (Mersilk 6-0, Ethicone, Raritan, NJ, USA) so that approximately 1/3–1/2 of the diameter of the nerve was trapped in the ligature [86]. The wound was closed with 4-0 silk sutures, and the animals were allowed to recover for one week.

Nociceptive threshold measurements were repeated on the 7th postoperative day for each animal at short intervals, and percentages of hyperalgesia/allodynia values were calculated with the following formula for the operated paws: hyperalgesia/allodynia (%) = 100 × (preoperative − postoperative values)/(preoperative values). Only animals that developed a minimum of 20% decrease of threshold with each method were used (*n* = 41). Rats were arranged into 5 groups having similar degrees of hyperalgesia/allodynia and received (1) i.t. solvent or (2) CART(55-76) or (3) CART(62-76) or (4) CART(55-76) 15 min after subcutaneous NTX pre-treatment or (5) CART(62-76) 15 min after s.c NTX administration. CART(55-76) and CART(62-76) were used in 3 nmol/rat and 10 nmol/rat concentrations, respectively. Measurements were carried out after i.t. injection starting with DPA at 20 min, followed by the Randall–Selitto test at 25 min, and finished with the noxious cold stimulation at 30 min. In the DPA test, rats were placed into a plexiglass chamber put on a metal mesh surface. Touch stimulator was positioned under the animal’s paw, and an increasing upward force (10 g/s) was exerted until the rat removed its paw. Withdrawal thresholds were measured 3 times in turns for each paw, and the mean values were used for statistical analysis. The pre-set maximum (50 g) was used in the evaluation if no withdrawal occurred. The Randall–Selitto test was performed as described in detail in the carrageenan-induced hyperalgesia section. To measure noxious cold sensitivity, hind paws of lightly restrained rats were immersed into a 0 °C water bath, and the latency to paw withdrawal was recorded. The cut-off time was set to 180 s.

For statistical comparisons of hyperalgesia/allodynia values, two-way ANOVA followed by Bonferroni post hoc test were used, and *p* < 0.05 was considered statistically significant.

### 4.6. Opioid Receptor Binding Assay

#### 4.6.1. Preparation of Brain Samples for Binding Assays

Four rats and three guinea pigs were decapitated, and their brains were quickly removed. The brains were prepared for membrane preparation according to Benyhe et al. [88], and part were used for binding experiments and part were further prepared for the [^35^S]GTPγS binding experiments according to Zador et al. [89].

The brains were homogenized in 30 volumes (*v/w*) of ice-cold 50 mM Tris-HCl pH 7.4 buffer with a Teflon-glass Braun homogenizer operating at 1500 rpm. The homogenate was centrifuged at 18,000 rpm for 20 min at 4 °C, the resulting supernatant was discarded, and the pellet was taken up in the original volume of Tris-HCl buffer. The homogenate was incubated at 37 °C for 30 min in a shaking water-bath. Then, centrifugation was repeated as described before. The final pellet was suspended in 5 volumes of 50 mM Tris-HCl pH 7.4 buffer containing 0.32 M sucrose, stored at −80 °C.

For the [^35^S]GTPγS binding experiments, the brains were homogenized with a Dounce in 5 volumes (*v/w*) of ice-cold TEM (Tris-HCl, EGTA, MgCl_2_) and stored at −80 °C. The protein content of the membrane preparation was determined by the method of Bradford, with BSA being used as a standard [90].

#### 4.6.2. Functional [^35^S]GTPγS Binding Experiments

In [^35^S]GTPγS binding experiments, the GDP→GTP exchange of the G_αi/o_ protein was measured in the presence of a given ligand to determine the potency of the ligand and the maximal efficacy of the receptor G-protein. The nucleotide exchange was monitored by a radioactive, non-hydrolysable GTP analogue, [^35^S]GTPγS.

The functional [^35^S]GTPγS binding experiments were performed as previously described [91,92] with modifications. Briefly, the membrane homogenates were incubated at 30 °C for 60 min in Tris-EGTA buffer (pH 7.4) composed of 50 mM Tris-HCl, 1 mM EGTA, 3 mM MgCl_2_, 100 mM NaCl, containing 20 MBq/0.05 cm^3^ [^35^S]GTPγS (0.05 nM), and increasing concentrations (10^−10^–10^−5^ M) of CART peptides and DAMGO. The experiments were performed in the presence of excess GDP (30 μM) in a final volume of 1 mL. Total binding was measured in the absence of test compounds, and non-specific binding was determined in the presence of 10 μM unlabelled GTPγS and subtracted from the total binding. The difference represented the basal activity. The reaction was terminated by rapid filtration under vacuum (Brandel M24R Cell Harvester, Alpha Biotech, Glasgow, UK) and washed three times with 5 mL ice-cold 50 mM Tris-HCl (pH 7.4) buffer through Whatmann GF/B glass fibres. The radioactivity of the dried filters was detected in an UltimaGold^TM^ MV aqueous scintillation cocktail with Packard Tricarb 2300TR liquid scintillation counter. [^35^S]GTPγS binding experiments were performed in triplicates and repeated at least three times.

#### 4.6.3. Competitive Binding Experiments

In competitive binding experiments, we used radio-labelled ligands in increasing concentrations and measured the amount of specifically bound radioactive ligands in the function of the applied radio-ligand concentrations.

Aliquots of frozen rat and guinea pig brain membrane homogenates were centrifuged (20 min, 18,000 rpm, 4 °C) to remove sucrose, and the pellets were suspended in 50 mM Tris-HCl buffer (pH 7.4). Membranes were incubated in the presence of the unlabelled DAMGO, HS665, and CART peptides in increasing concentrations (10^−10^–10^−5^ M) at 35 °C for 45 min with [^3^H]DAMGO and 30 °C for 45 min with [^3^H]HS665. The non-specific and total binding were determined in the presence and absence of 10 µM unlabelled naloxone and HS665, respectively. The reaction was terminated by rapid filtration under vacuum (Brandel M24R Cell Harvester) and washed three times with 5 mL ice-cold 50 mM Tris-HCl (pH 7.4) buffer through Whatman GF/C glass fibres. The radioactivity of the dried filters was detected in an UltimaGold^TM^ MV aqueous scintillation cocktail with Packard Tricarb 2300TR liquid scintillation counter. The competitive binding assays were performed in duplicate and repeated at least three times.

### 4.7. DPP4 Inhibitor Screening Assay

To evaluate the effect of the CART(55-76) fragment on DPP4 enzyme activity, the commercially available DPP4 Inhibitor Screening Assay Kit (Abcam, Cambridge, UK, ab133081) was used, and the protocol provided by the manufacturer was followed. The assay used the fluorogenic substrate, Gly-Pro-Aminomethylcoumarin (AMC), and following the cleavage of the substrate, fluorescence of the released AMC group was analysed with an excitation wavelength of 355 nm and an emission wavelength of 460 nm. Background fluorescence was measured in three wells containing fluorogenic substrate in assay buffer. Further three wells with DPP4 enzyme and substrate dissolved in assay buffer were used to determine the 100% enzyme activity. Three more wells contained the reference inhibitor sitagliptin beside DDP4 and the substrate. Logarithmic dilutions of CART(55-76) and IPI were assayed in triplicates. Curves were fitted on data using the non-linear regression function of the GraphPad Prism 5 software (GraphPad Software Inc., San Diego, CA, USA).

### 4.8. Mass Spectrometry

#### 4.8.1. Cerebrospinal Fluid Collection and Peptide Extraction

Cerebrospinal fluid (CSF) was collected as described previously [93]. Briefly, under deep anaesthesia, heads of rats were flexed approximately 45 degrees downward and, without making any incision, a butterfly needle connected to a 1 mL syringe was punctured into the cisterna magna. Non-contaminated samples of 100–200 µL CSF were taken from 3 naïve rats (NC11, NC12, and NC13), 3 rats with unilateral carrageenan-induced hind paw inflammation (GC20, GC21, and GC22), and 3 inflamed rats that received CART(55-76) peptide intrathecally (CC31, CC32, and CC33). A total of 50 µL CSF from each rat was mixed with 250 µL of cold acetonitrile (4 °C), vortexed thoroughly, and centrifuged for 10 min at 14,000 rpm. Supernatants were collected, transferred to a clean vial, and kept at −80 °C until measurement.

#### 4.8.2. Peptide Extraction from Spinal Cord Segments

Three naïve adult male Wistar rats (NG10, NG11, and NG12), 3 rats with unilateral carrageenan-induced hind paw inflammation (GG9, GG10, and GG11), and 3 inflamed rats that received CART(55-76) peptide intrathecally (CG9, CG10, and CG11) were decapitated, and L3–L5 spinal cord segments were removed and snap frozen on dry ice. Tissue was homogenized by sonication in cold acetonitrile (4 °C) in a volume of 10 µL/mg of tissue and centrifuged for 10 min at 14,000 rpm. Supernatant was transferred into Eppendorf tubes and stored at −80 °C until use.

#### 4.8.3. Measuring the Concentration of IPI

Samples were injected directly as obtained from the extraction protocols. Five parallel injections were acquired from each sample to see the accuracy of the measurements. Standard solution of IPI was used for making the calibration curve in the range of 0–50 ng/mL. Experimental quantification of IPI was taken on a Sciex 6500QTRAP hybrid tandem mass spectrometer (Framingham, MA, USA) equipped with TurboV ion source. Separation was performed on a Perkin Elmer Series200 micro HPLC system (Waltham, MA, USA) consisting of a binary pump, an autosampler, and a column oven. A Merck Purospher Star C18 (55 mm × 2 mm, 3 µm) column was used for separating the IPI from the matrix component. Water containing 0.1% formic acid (eluent A) and acetonitrile containing 0.1% formic acid (eluent B) were used for separation in gradient mode. The initial concentration (A/B) was 95/5 and was kept for 2 min. A linear gradient was used for 4 min to obtain the 5/95 composition, and this was kept for 2 min. The initial concentration was reached within 0.5 min, and the equilibration time was 5.5 min. The flow rate was 0.5 mL/minute. Column temperature was ambient. The injection volume was 10 µL. Electrospray ionization was used for detection in positive MRM mode. MRM transitions were identified in product ion scan mode. Q1/Q3 masses were the following: quantifier transition of 342/229 and qualifier transition of 342/70. Dwell times were 150 ms for each transition. Source parameters were the following: curtain gas of 45 au (arbitrary unit), spray voltage of 5000 V, source temperature of 450 °C, nebulizing gas of 35 au, drying gas of 40 au, entrance potential of 10 V, collision energy of 30 V, and CXP of 15 V. Analyst version 1.6.3 was used for instrument control and data processing.

### 4.9. Cell Culture Experiments

#### 4.9.1. Primary Astrocyte Cultures

For the production of spinal cord astrocyte cultures, we followed the procedure already described by Pap et al. [94] with slight modifications. Briefly, the whole spinal cord was removed from 7–8 day-old Wistar rat pups after decapitation and was placed into ice-cold dissecting buffer (136 mM NaCl, 5.2 mM KCl, 0.64 mM Na_2_HPO_4_, 0.22 mM KH_2_PO_4_, 16.6 mM glucose, 22 mM sucrose, 10 mM HEPES supplemented with 0.06 U/mL penicillin and 0.06 U/mL streptomycin). The isolated spinal cords were cleaned of the meninges, then were placed into fresh dissecting solution containing 0.025 g/mL bovine trypsin (Merck–Sigma-Aldrich), and were incubated at 37 °C for 30 min. Then, the solution was replaced by Minimum Essential Medium (MEM, Gibco, Life Technologies Ltd., Parsely, UK) supplemented with 10% Fetal Bovine Serum (FBS, Hyclone, GE Healthcare Bio-Sciences, Pittsburgh, PA, USA). Finally, cells were suspended by gentle suspension of the tissue pieces by a Pasteur pipette. The cell number was identified, the cell density was set to 1 × 10^6^/mL, and the cells were placed into 24 well tissue culture plates (0.5 mL/well). The cell cultures were kept at 37 °C in a 5% CO_2_ atmosphere, and the medium was replaced the following day and then every second day. On the 7th day of cell culturing, the loosely attaching microglial cells were removed by shaking the culture plates at 200 rpm on a rotary shaker at 37 °C for 30 min. Then, between 8 and 10 days of culturing, the cells were used for further experiments.

#### 4.9.2. Immunocytochemistry

Immunocytochemistry was performed on astrocyte cultures, which were kept on coverslips placed into 24 well culture dishes. After 10–12 days of culturing, the coverslips were removed, and the cells were placed into PBS. Some of the cell cultures were treated with 1 μg/mL lipopolysaccharide (LPS, Merck–Sigma-Aldrich) for 6 h. The cell cultures were fixed with 4% paraformadehyde (15 min). Then, the cells were washed in PBS containing 100 mM glycine followed by 10 min incubation in PBS. Non-specific labelling was blocked by PBS containing 10% normal serum for 50 min. Then, the cells were incubated with the primary antibodies (goat-anti-DPP4 [33], mouse-anti-TLR4 [95]; rabbit-anti-caveolin-1, 1:500 [96]; guinea pig-anti-GFAP [97]; for details see Appendix A) overnight at 4 °C. The following day, the cells were incubated with the appropriate secondary antibodies (120 min, RT). Finally, the cell nuclei were stained with DAPI. Immunofluorescent images were acquired by an Olympus FV3000 confocal microscope with a 60× oil-immersion lens (NA: 1.4). Single 1-μm-thick optical sections were scanned from the cell cultures, and the confocal settings (laser power, confocal aperture and gain) were identical for all methods. The scanned images were processed by Adobe Photoshop CS5 software.

#### 4.9.3. Cell Treatments

10–12-day-old astrocyte cultures were stimulated with 1 μg/mL lipopolysaccharide for 4–16 h [38]. Some of the cultures received 0.1 or 1 μM Ile-Pro-Ile treatment for an additional 3 h. Then, the culture supernatants were removed. Cells attaching the culture plates were lysed, and cytosol and nuclear fractions were separated as was already described [98]. Supernatants, cytosol, and nuclear fractions were kept in aliquots at −70 °C until they were used.

#### 4.9.4. Western Blot Analysis of Astrocytic DPP4 Expression

Fractions of 50 µg protein/lane cytosol were run on 10% polyacrylamide gels. The separated proteins were electrophoretically transferred onto PVDF membranes. Membrane blocking in 5% bovine serum albumin (BSA, 2 h, RT) was followed by overnight incubation with goat anti-DPP4 (1:1000, Novus Biological–Bio-Techne, Minneapolis, MN, USA) and mouse anti-β-tubulin (loading control, 1:2000, Merck-Sigma-Aldrich) primary antibodies at 4 °C. Then, the membranes were treated by HRP conjugated rabbit-anti-goat or rabbit-anti-mouse secondary antibodies (1:1000, DAKO–Agilent, Santa Clara, CA, USA), respectively. Finally, colour reactions were developed by 3,3′-diaminobenzidine or aminoethyl-carbasole chromogens. Relative protein levels for DPP4 upon 16 h of LPS treatment were quantified by densitometry in three independent astrocyte cultures and analysed by the Kolmogorov-Smirnov statistical test.

#### 4.9.5. Enzyme Linked Immunosorbent Assay (ELISA)

ELISA was performed as already described [99]. Briefly, (after titration in increasing dilutions) the astrocyte culture supernatants were diluted in a 1:1 ratio by ELISA coating buffer (15 mM Na_2_CO_3_, 35 mM NaHCO_3_, pH 9.6) and were used for the coating of 96 well polystyrene ELISA plates (Maxisorp, NUNC Intermed, Copenhagen, Denmark). Free binding capacity of the wells was blocked by 1% gelatin (Reanal, Budapest, Hungary) in PBS. After washing with PBS, anti-IL-6 [98] and anti-TNFα [100] primary antibodies were added (2 h, 37 °C), followed by goat-anti-rabbit-HRP secondary antibodies (1:2000, DakoCytomation, Glostrup, Denmark). Colour reaction was developed by o-phenylene-diamine, and the optical density was measured at λ = 492 nm by ELISA plate reader (Titertek, Uniscan, Flow Laboratories, Helsinki, Finland).

Optical density data were averaged, and the standard error of mean values was calculated for each treatment. Bar charts showed relative changes in the expression level of cytokines in percentages, and the control level was set to 100%. Graphs show representative data out of three independent experiments. Statistically significant differences were calculated by one-way ANOVA followed by Student-Newman-Keuls pairwise comparison. The difference between groups was considered statistically different if *p* < 0.05.

### 4.10. Statistical Analysis

Statistical methods used are detailed at each experiment individually. Analyses were made and curves/bar graphs were created with the GraphPad Prism 5 software.

## Figures and Tables

**Figure 1 ijms-24-00918-f001:**
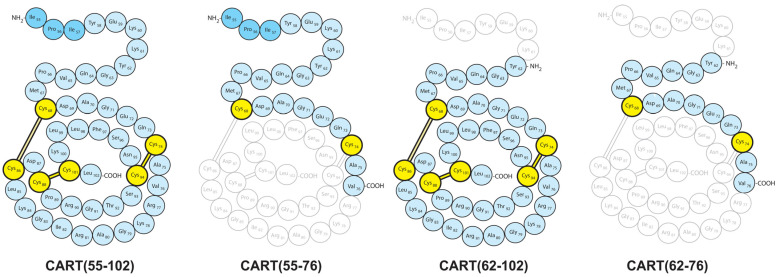
Schematic diagram of CART fragments (based on the original publication of Thim et al. [7]).

**Figure 2 ijms-24-00918-f002:**
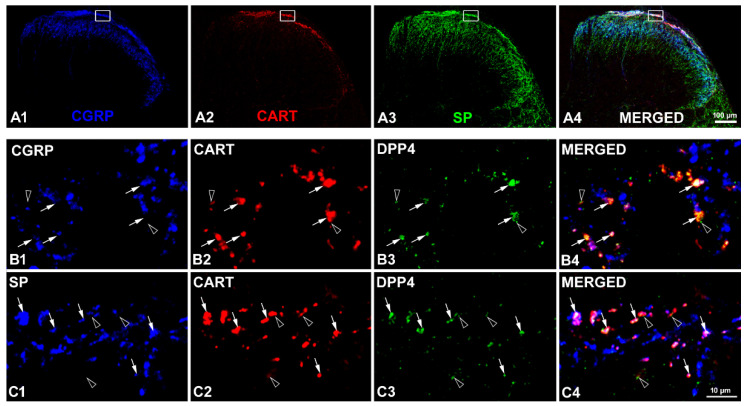
Coexistence of CART peptide and DPP4 in axon terminals in the rat spinal dorsal horn. Triple immunofluorescent labelling with CGRP, CART, and SP antibodies demonstrates the landmarks of the rat spinal dorsal horn (**A1**–**A4**) and shows the approximate location of the region of interest (white rectangle). DPP4 immunoreactivity (green) is strongly associated with CART-positive (red) peptidergic C primary afferent terminals also containing CGRP (blue) in (**B1**–**B4**) or with CART-ergic primary afferents and local interneurons expressing SP (blue) in (**C1**–**C4**). Images (**B4**,**C4**) were merged from images (**B1**–**B3**,**C1**–**C3**), respectively. All the high-power images are single optical sections (Immunofluorescent labelling was performed on three naïve rats. Confocal images of any randomly selected fields, sections, and rats showed consistent results). Arrows: co-localization of CART and DPP4 in axon terminals, empty arrowheads: close proximity of CART immunoreactive terminals and DPP4. Scale bars: 100 µm (**A1**–**A4**) and 10 µm (**B1**–**B4**,**C1**–**C4**).

**Figure 3 ijms-24-00918-f003:**
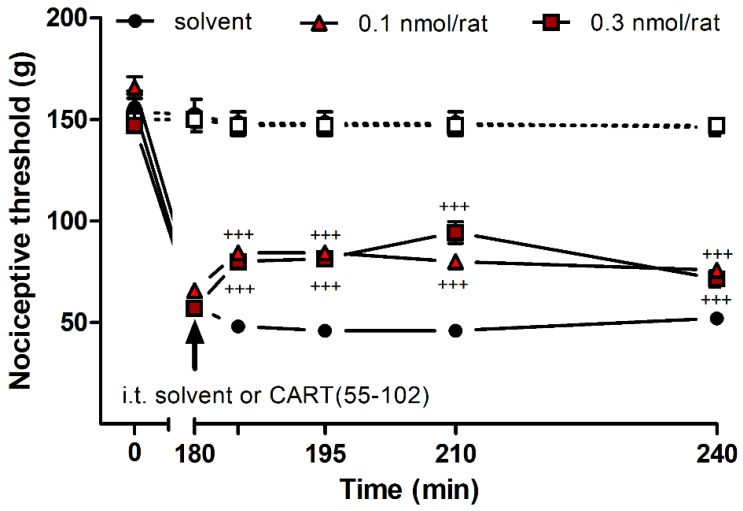
Antihyperalgesic effect of CART(55-102) in carrageenan-induced inflammation determined by the Randall–Selitto method. I.t. CART(55-102) had a significant antihyperalgesic effect in 0.1–0.3 nmol/rat concentrations (two-way ANOVA, Bonferroni post hoc test. ^+++^: *p* < 0.001), but higher doses resulted in an ascending substantial tremor, making the evaluation of the test unreliable. CART(55-102) never altered the nociceptive threshold of the contralateral hind paw significantly. Five rats were involved in the vehicle group, while seven animals were used for each drug concentration. Filled symbols: side of inflammation; open symbols with dashed lines: contralateral side. Data on each curve are given as mean ± standard error of the mean (SEM).

**Figure 4 ijms-24-00918-f004:**
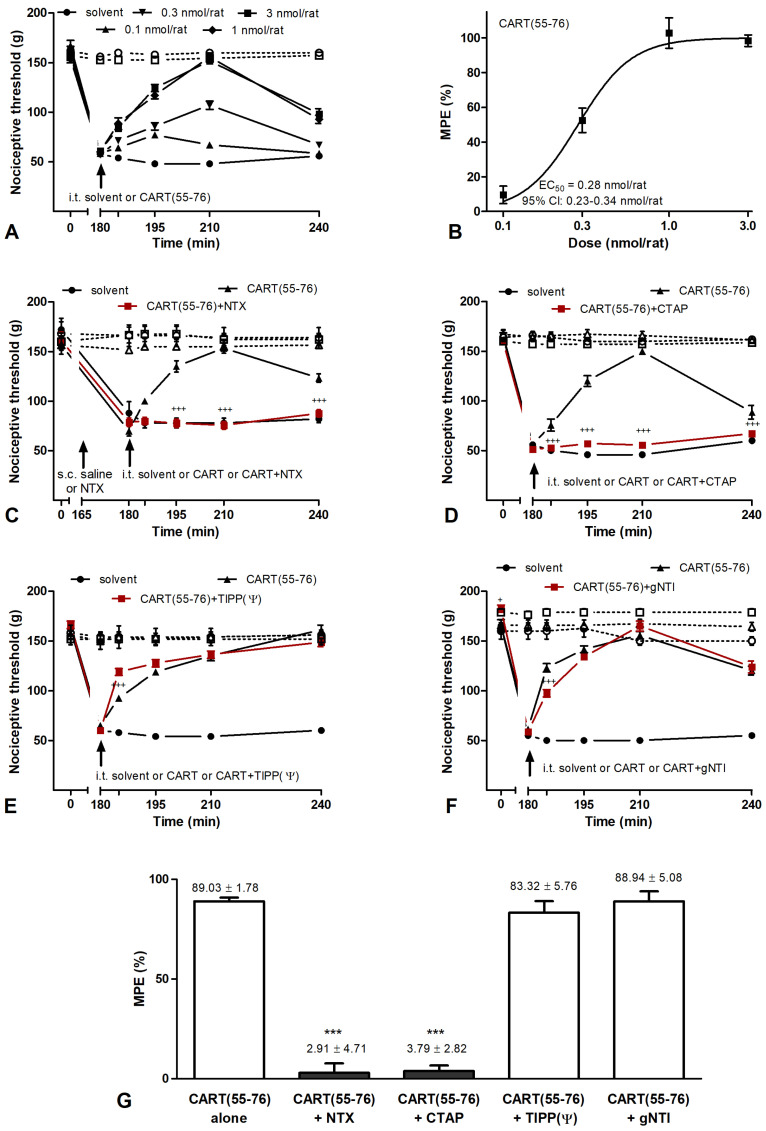
Antihyperalgesic effect of CART(55-76) peptide is µ-opioid receptor-dependent in the carrageenan-induced hyperalgesia model measured by the Randall–Selitto method. (**A**) I.t. CART(55-76) had a dose-dependent antihyperalgesic effect without causing motor abnormalities. (**B**) A sigmoidal dose–response curve of CART(55-76) was plotted onto data obtained from the nociceptive threshold values recorded at 210 min. Antihyperalgesic ED50 value of CART(55-76) was calculated as 0.28 (0.23–0.34) nmol/rat, given as mean and 95% confidence interval. (**C**) Antihyperalgesic effect of CART(55-76) (3 nmol/rat, i.t.) was completely abolished by the non-selective opioid antagonist NTX (0.5 mg/g b.w., s.c.) and (**D**) the µ-selective opioid antagonist CTAP (200 pmol/rat, i.t.). (**E**) Neither the δ-antagonist TIPP(Ψ) (1 nmol/rat, i.t.) nor (**F**) the κ-selective antagonist gNTI (10 nmol/rat, i.t.) had an influence on the antihyperalgesic effect of CART(55-76). CART(55-76) never altered the nociceptive threshold of the contralateral hind paw significantly, and none of the antagonists tested had a significant effect when they were injected alone. (**G**) Inhibitory effects of opioid antagonists on CART(55-76)-related antihyperalgesia are summarized on the bar graph created from values measured at 210 min after i.t. drug administration (MPE %: maximal possible effects in percentage). Five rats were involved in each vehicle group, while 7–9 animals were used for each drug and drug combination/concentration. Comparisons were made with two-way ANOVA, Bonferroni post hoc test; +++: *p* < 0.001 (**C**–**F**), and one-way ANOVA followed by Dunnett’s post hoc test. ***: *p* < 0.001 (**G**). Crosses always indicate a significant difference between the time-matching points of CART(55-76) and CART(55-76) + antagonist curves. Filled symbols: side of inflammation; open symbols with dashed lines: contralateral side. Data on each curve and bar are given as mean ± SEM.

**Figure 5 ijms-24-00918-f005:**
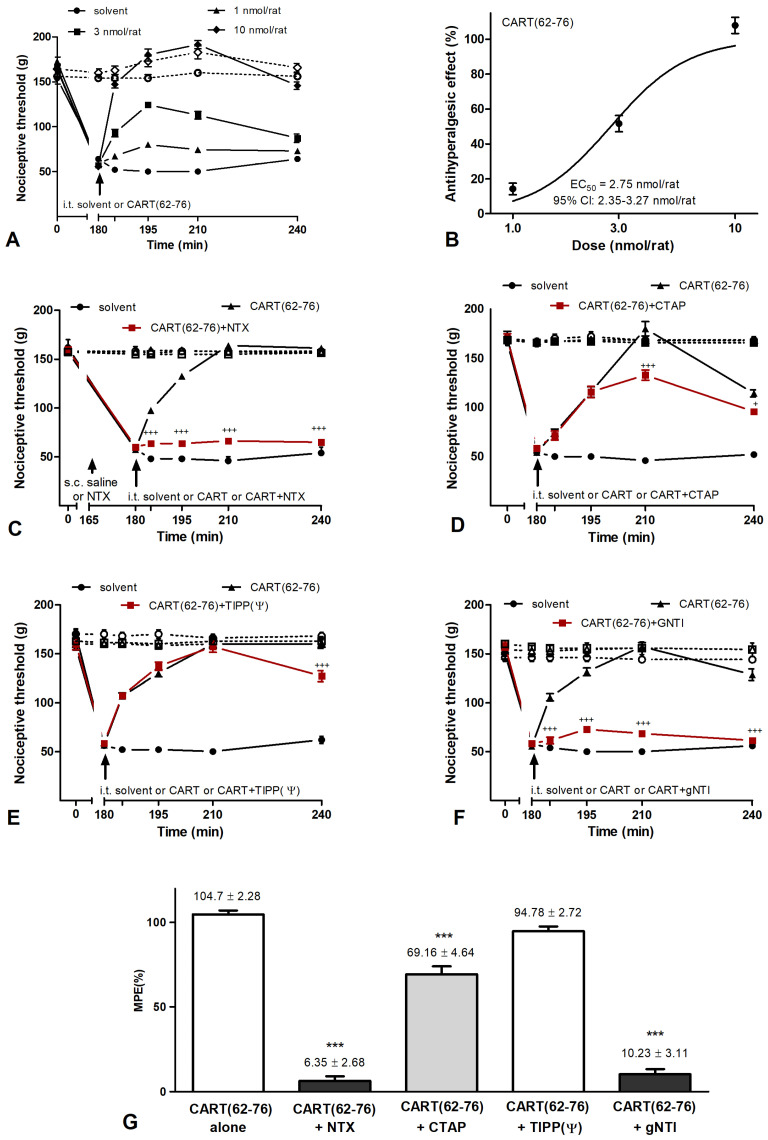
Antihyperalgesic effect of CART(62-76) peptide is mainly κ-opioid receptor-dependent in carrageenan-induced subacute inflammation. (**A**) I.t. CART(62-76) (10 nmol/rat i.t.) had a significant antihyperalgesic effect in 3–10 nmol/rat concentrations without any signs of motor disfunctions. (**B**) Dose–response curve was constructed from nociceptive threshold values recorded at 210 min. Antihyperalgesic ED50 value of CART(62-76) was calculated as 2.75 (2.32–3.27) nmol/rat, given as mean and 95% confidence interval. (**C**) Subcutaneous pre-treatment with the non-selective opioid-receptor antagonist NTX (0.5 mg/g b.w., s.c.) completely eliminated the antihyperalgesic effect of CART(62-76). (**D**) The µ-receptor selective antagonist CTAP (200 pmol/rat, i.t.) slightly reduced the antihyperalgesic effect of CART(62-76). (**E**) δ-opioid selective antagonist TIPP(Ψ) (1 nmol/rat, i.t.) did not alter the antihyperalgesic effect of CART(62-76). (**F**) κ-selective antagonist gNTI (10 nmol/rat, i.t.) co-injected i.t. completely abolished the antihyperalgesic effect of CART(62-76). Apart from only one set of experiments, CART(62-76) never altered the nociceptive threshold of the contralateral hind paw significantly, and none of the antagonists tested had a significant effect when they were injected alone. (**G**) Bar graph summarizes inhibitory effects of opioid antagonists on CART(62-76) related antihyperalgesia. Five rats were involved in each vehicle group, while 7–9 animals were used for each drug and drug combination/concentration. Comparisons were made with two-way ANOVA, Bonferroni post hoc test; +: *p* < 0.05; +++: *p* < 0.001 (**C**–**F**), and one-way ANOVA followed by Dunnett’s post hoc test. ***: *p* < 0.001 (**G**). Crosses always indicate a significant difference between the time-matching points of CART(62-76) and CART(62-76) + antagonist curves. Filled symbols: side of inflammation; open symbols with dashed lines: contralateral side. Data on each curve and bar are given as mean and SEM.

**Figure 6 ijms-24-00918-f006:**
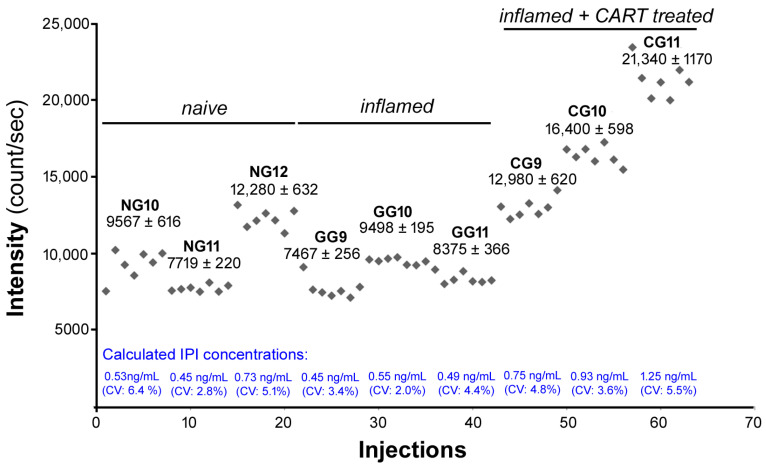
IPI was cleaved immediately from exogenously applied CART(55-76) peptides under inflammatory conditions. The distribution of peak intensities and calculated IPI concentrations of spinal segments were measured by mass spectrometry. Peak intensities were determined in seven parallel runs, and data are given as counts/second, mean ± standard deviation (SD), (CV%). Calculated IPI concentrations (mean, coefficient of variance %) were analysed with one-way ANOVA: *p* ˂ 0.05, F_2,6_ = 6.88, *p* = 0.028, all pairwise multiple comparison procedure, Student-Newman-Keuls post hoc test; *p* (inflamed vs. inflamed + CART) = 0.032, *p* (inflamed + CART vs. naive) = 0.027, *p* (naive vs. inflamed) = 0.0618.

**Figure 7 ijms-24-00918-f007:**
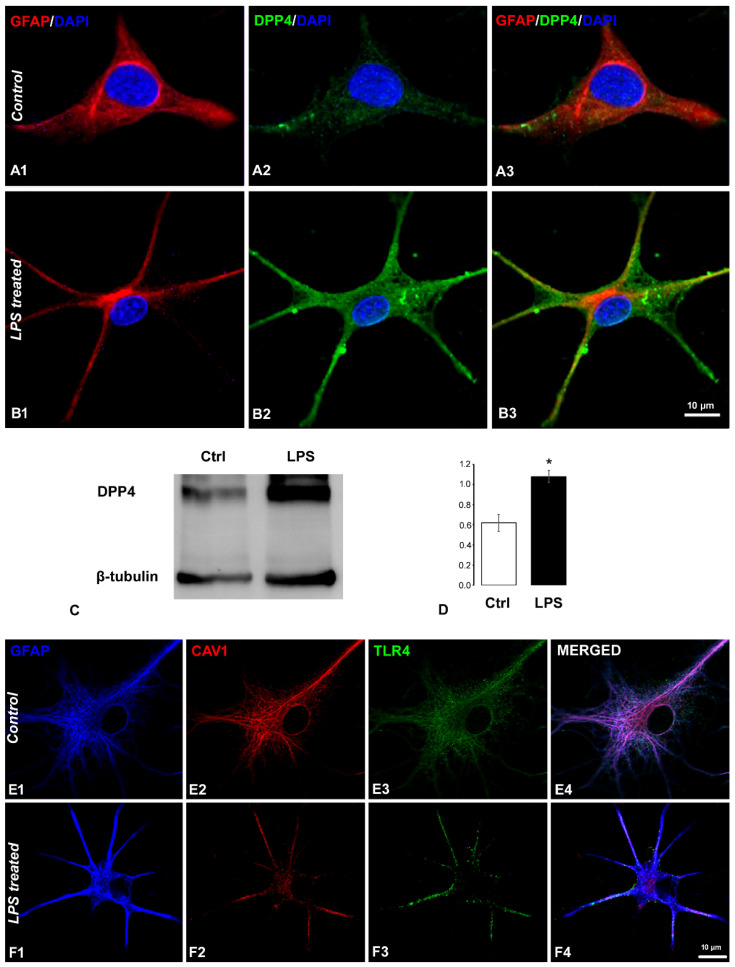
DPP4, TLR4, and caveolin-1 expression was enhanced upon LPS stimulation in cultured rat spinal astrocytes. (**A**,**B**) Fluorescent micrograph images illustrating co-localization between the GFAP astrocytic marker ((**A1**,**B1**) red) and DPP4 ((**A2**,**B2**) green). Mixed colors (yellow) on the superimposed images (**A3**,**B3**) indicate double labelled structures. On all images, 4’,6-diamidino-2-phenylindole (DAPI) was used to label cell nuclei (blue). (**C**) Representative immunoblot also shows increased expression of DPP4. (**D**) Densitometric analysis of western blots shows significant overexpression of the DPP4 protein after 16 h of LPS treatment. Results are shown as average ± SEM of three independent astrocyte cultures. (* *p* = 0.03262, Kolmogorov-Smirnov statistical test). (**E**,**F**) Representative confocal images illustrate TLR4 (green) and caveolin-1 (red) expression in GFAP-immunoreactive astrocytes (blue). In (**E1**–**E4**) images, the control culture is seen. In the case of LPS-treated astrocytes (**F1**–**F4**), characeristic rearrangement of TLR4 and caveolin-1 subcellular localization was observed. Scale bar: 10 μm.

**Figure 8 ijms-24-00918-f008:**
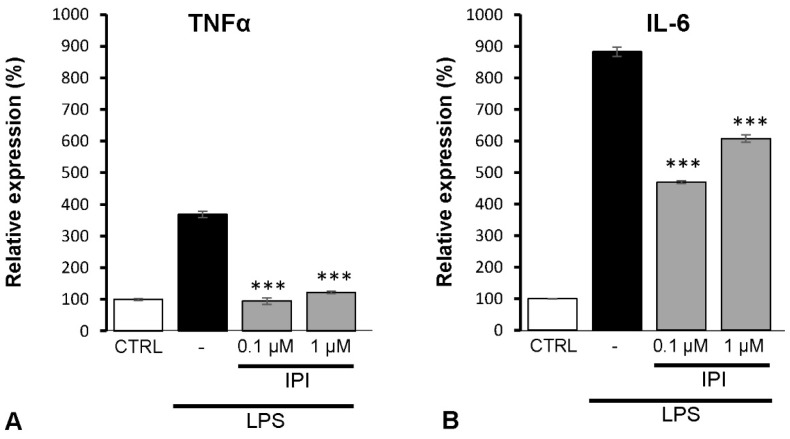
The DPP4 inhibitor Ile-Pro-Ile peptide (IPI) significantly attenuates LPS-induced TNFα and the IL-6 secretion of spinal astrocytes. Histograms show relative TNFα (**A**) and IL-6 (**B**) levels determined by the ELISA method after 4 h of pre-stimulation by LPS followed by 3 h of IPI treatment. In the case of LPS-pre-stimulated cultures, both TNFα and IL-6 secretion of the astrocyte cultures were significantly down regulated by 0.1 or 1 µM IPI if compared with the LPS-treated cultures. Data are shown as mean ± SEM of a representative ELISA out of three independent experiments. One-way ANOVA, followed by Student-Newman-Keuls pairwise comparison, *** *p* < 0.001.

## Data Availability

The datasets generated and analysed during the current study are available from the corresponding author on request.

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
