# Peer review of "Unique, Specific CART Receptor-Independent Regulatory Mechanism of CART(55-102) Peptide in Spinal Nociceptive Transmission and Its Relation to Dipeptidyl-Peptidase 4 (DDP4)"

_ijms, 2023, doi:10.3390/ijms24020918_

Round 1

Reviewer 1 Report

Kozsurek et al. examine the effects of cocaine- and amphetamine-regulated transcript (CART) peptides on nociceptive transmission in the rat spinal dorsal horn by using various techniques such as immunofluorescence, Western blot, behavioral technique (Randall-Selitto and tail-flick tests), mass spectroscopy, binding assay and ELISA.  As a result, they found out (1) a coexistence of CART peptide and DPP4 (dipeptidyl-peptidase 4) in CGRP- or SP-containing C-primary afferent terminals in the rat spinal dorsal horn, (2) a localization of CART (55-102) and CART (62-102) in the rat spinal cord, (3) a CART (55-102)-induced inhibition of carrageenan-induced inflammatory pain, (4) a CART (55-76)-induced inhibition of carrageenan-induced inflammatory pain in a manner dependent on μ-opioid receptors, (5) a CART (62-76)-induced inhibition of carrageenan-induced subacute inflammatory pain in a manner dependent on μ- and κ-opioid receptors and (6) a CART (55-76)- and CART (62-76)-induced inhibition of neuropathic pain produced by partial sciatic nerve ligation.  Moreover, they demonstrated (1) no affinity of CART (55-76) and CART (62-76) to μ- and κ-opioid receptors in rat and guinea-pig brain membrane homogenates, (2) a cleavage of isoleucine-proline-isoleucine (IPI; DPP4 inhibitor) from CART (55-76) in carrageenan-induced inflamed rat spinal cord, (3) enhanced DPP4 expression by lipopolysaccharide (LPS) treatment in cultured rat spinal astrocytes, (4) an IPI-induced inhibition of LPS-induced TNFα and IL-6 secretion from cultured rat spinal astrocytes and (5) enhanced TLR4 and caveolin-1 expression in cell surface by LPS treatment in cultured rat spinal astrocytes.  As a result, it was concluded that when inflammation occurs, CART (55-102) is processed in the spinal cord, leading to the production of IPI that binds to DPP4, which in turn leads to a decreased level of TNFα and IL-6 and thus to analgesia.  This manuscript does not seem to have been written carefully in presentation and scientific point of view and thus is a little difficult to read.  There are so many points that should be addressed and may be useful to amend this manuscript, as follows:

Major points:

1.     Title: this should be revised, because it is unknown from this title what kind of receptors is independent of regulatory mechanism and in which region the pain transmission occurs.

2.     Abstract: this does not seem to appropriately summarize the results obtained in the present study.  For example, there is no description about an involvement of opioid receptors, as shown in Figs. 5 and 7.  Although the expression of TLR4 in astrocytes is shown in Fig. 13, it is not shown that DPP4 interacts with TLR4 (see lines 58 and 59).  The abstract should be revised.

3.     Fig. 2: how many times the result shown in this figure was confirmed?  Please reply to this question.

4.     Fig. 3: was the densitometric analysis of this result performed?  Please reply to this question.

5.     Pages 5 and 9: each of Fig. 5A-G, 7A-G and 8A-F should be stated separately.

6.     The third paragraph on page 5: it is unknown what NTX, CTAP, TIPP(Ψ) and gNTI mean without reference of page 17 (Materials and Methods).  It may be better to give a list of abbreviations.  The concentrations of the drugs used should be given in the legends of Figs. 5 and 7, even though these values are mentioned in Materials and Methods.  Although the authors mention that TIPP(Ψ) and gNTI are ineffective in lines 187-189, is it possible that the concentrations used were not sufficient?  What is the SEM of 2.75 (line 235) in Fig. 7B?  Please reply to these questions.

7.     Fig. 7B: the curve fitting of data with three different concentrations would be inappropriate.  Experiments with more concentrations should be performed.

8.     Figure legends: enough information does not seem to be given in several of them.  It should be stated how many times the experiment was performed in each legend, even though this number is mentioned in Materials and Methods.

9.     The first part of the figure legend shown in bold should state the result obtained from the figure.  Please amend the legends of Figs. 3, 6, 9 and 10.  The first parts of the legends of Figs. 5 and 7 should mention an involvement of opioid receptors.

10.  Fig. 9: in this experiment, the brain but not spinal cord is examined.  Why the brain was studied?  Please reply to this question.

11.  Lines 358-360: it would be better for the authors to give as a supplemental material data showing that intraplantar injection of carrageenan induces TLR4 and CAV1 expression in the spinal cord, because this data is used in Discussion (lines 454 and 455).

12.  The second paragraph on page 14: the present study does not give specific data about motor effects.  Therefore, description of motor effects should be omitted.  It is not sure whether the authors’ findings are related to gliogenic LTP.  Therefore, “gliogenic LTP” should be deleted.  Moreover, “gliogenic LTP” should be omitted in the second paragraph on page 16 as well.

13.  Lines 411-415: was the result about “ascending tremor” mentioned here obtained in the present study?  If so, this observation should be stated in a quantitative manner in Results.  If this observation has been reported previously, please cite the literature that reported this result.

14.  Lines 437-439: it is not clear why the enhancement by DPP4 of LPS-induced IL-6 and TNFα expression reveals an interaction between TLR4 and DPP4.  Please make this point clear.

15.  Lines 448 and 449: please explain “CUX1” shortly.  What is “this study”?  These sentences should be rewritten for clarity.

Specific points:

1.     Lines 65, 386, 489, 502 and so on: does “CNS” mean an involvement of not only the spinal cord but also the brain in the authors’ findings?  Please reply to this question.

2.     Line 75: please put a period following “[4-6]”; not “structure” but “structures”?

3.     Line 111: please put a period following “(hyperalgesia)”.

4.     Lines 135, 331 and 371: not “spinal” but “rat spinal”?

5.     Line 137: not “raw” but “row”?

6.     Line 171: please define “SEM”.  Please use either “SEM” or “S.E.M.” (line 292) throughout the text.

7.     Line 210: what is the SEM of 0.28?  Please reply to this question.

8.     Line 253: please give a literature in “as stated previously”.

9.     Fig. 9: is the base of logarithm in Figs. 9A-C “10”?  If so, please amend this figure.  “g” in Fig. 9C should be “γ”.

10.  Line 312: please expand “SD”.

11.  Line 334: please define “DAPI”.

12.  Line 336: not “show” but “shows”.

13.  Fig. 12: please put a space between μM and the number (0.1 or 1).

14.  Line 364: please define “GFAP”.

15.  Line 377: “These are ..” is inappropriate expression.  Please amend this point.

16.  Line 400: please use “CNS” here.

17.  Table 1: please give the full characters of “Specie” and “Dilutio”.

18.  In all figure legends, its first part should be shown in bold.

19.  Lines 573, 605 and 612: please use either “minutes”, “min” or “mins” throughout the text.

20.  Line 630: does not “Euthasol” contain phenytoin?   Please reply to this question.

21.  Line 661: please put “et al.” following “Benyhe”.

22.  Line 663: please put “et al.” following “Zador”.

23.  Line 685: is “30 M” OK?  Please reply to this question.

24.  Line 687: is “10 M” OK?  Please reply to this question.

25.  Line 703: is “10 M” OK?  Please reply to this question.

26.  Line 757: is “was ambient.” OK?  Please correct English.

27.  Line 764: change “4.9. Cell ..” to a new line.

28.  Pages 21 and 22: please give information of the companies from which devices such as scintillation counter, mass spectroscopy and HPLC system were purchased.

29.  References: the style (in terms of journal name and title) of references given here is not unified; for example, please compare 6 and 7; 18 and 19.  Please amend this point.  Moreover, please check all of the references whether they are cited correctly.

30.  There appear to be more mistakes than pointed out above.  Please check the manuscript very carefully.

Author Response

Dear Editor and Reviewer 1,

We are very grateful for the stimulating comments and critics that we got in the review process. Please, find our point-by-point responses as an attached pdf file!

Sincerely yours,

Zita Puskar and Mark Kozsurek

Reviewer 2 Report

The subject of the study is interesting and relates to neuroinflammatory processes. The Authors use various in vivo and in vitro research models and methods. The Introduction and Discussion address the research thesis. The Authors propose new mechanisms for regulating neuroinflammatory processes, their research needs further confirmation. Discussion concise and to the point. The conclusions open up the discussion or could be included as a Summary. What is missing from the discussion is an attempt to explain why the effects of CART are so different.

The manuscript reads well, but some aspects need improvement.

1. The quality of the charts (Figure 5 and 7) is very poor e.g. no visible SEM indicators, the markers overlap. 

2. On Figure 5G and 7G the Authors present inhibitory effects of opioid antagonists on the specific CART-related antihyperalgesia, whereas the way of graph presentation suggest the own action of these compounds. Change the description of the x-axis is necessary. Were this compounds injected alone or after/before CART? - this information is missing in most compounds from the description of the graph.

3. CART(62-76) was ineffective alone but when was co-administered with NTX an antihyperalgesic effect was observed. Antihyperalgesic effect of CART(55-76) was completely abolished by the NTX. In Discussion the Authors suggests that NTX docking to TLR4 may affect the binding of DPP4 inhibitors to DPP4-CAV1 complex that cannot block the ongoing inflammatory processes, however the effect was opposite. Could these effects be related to the possible activity of NTX on the TLR4 receptor or rather to the CART specificity? 

4. Why were rats of both sexes and guinea pigs used in the binding study? Could this have influenced the binding results?

5. Figure 11. The photo B1-B3 is identical in details. It looks as if only the filter colours have been changed in the photos. Please check the photo and replace it with the correct one. 

6. In the discussion, it is worth expanding on why the effects of CART are so different (3.2).

7. line 764 - correct the position of the section, 756 - free spaces.

Author Response

Dear Editor and Reviewer 2,

We are very grateful for the stimulating comments and critics that we got in the review process. Please, find our point-by-point responses as an attached pdf file!

Sincerely yours,

Zita Puskar and Mark Kozsurek

Round 2

Reviewer 1 Report

This revised manuscript has been largely amended according to my comments, and there is no concern in the present manuscript except for the following minor comments:

1.     Line 54: not “]7]and” but “[7] and”.

2.     Lines 88 and 111: please use either “fibre” or “fiber” throughout the text; please see line 383.

3.     Line 123: in Fig. 2, A1-A4, scale bar = 100 μm.  Please make this point clear.

4.     Line 192: please put “receptor” following “mu-opioid”.

5.     Line 199: please use either “μ” or “mu” throughout the text; please see line 192.

6.     Line 204: not “is” but “are”.

7.     Line 215: please put “receptor” following “kappa-opioid”.

8.     Line 224: please use either “κ” or “kappa” throughout the text; please see line 215.

9.     Fig. 6: not “0.53ng/ml” but “0.53 ng/ml”.  Please put a space between value and unit throughout the text and figures.

10.  Lines 282 and 283: please use either “P” or “p” throughout the text.

11.  Line 293: “acidic fibrillary” should be “fibrillary acidic”.

12.  Line 331: not “is” but “are”?

13.  Line 496: “raise” should be “raises”.

14.  Line 554: not “7,5” but “7.5”?

15.  Line 708: is “10 M” OK?  Please check this concentration.

16.  Line 817: not “2h” but “2 hours”.

Author Response

Dear Reviewer 1,

We are very grateful for your time and for revising our resubmitted manuscript. All the recommended changes were done and labelled with blue colour.

Sincerely yours,

Zita Puskar and Mark Kozsurek